# Morphometric Assessment for Functional Evaluation of Coronary Stenosis with Optical Coherence Tomography and the Optical Flow Ratio in a Vessel with Single Stenosis

**DOI:** 10.3390/jcm11175198

**Published:** 2022-09-02

**Authors:** Yuming Huang, Zehuo Lin, Quanmin Wu, Liansheng Chen, Junqing Yang, Huiliang Deng, Yuanhui Liu, Nianjin Xie

**Affiliations:** 1Department of Catheterization Lab, Guangdong Cardiovascular Institute, Guangdong Provincial Key Laboratory of South China Structural Heart Disease, Guangdong Provincial People’s Hospital, Guangdong Academy of Medical Sciences, Guangzhou 510080, China; 2Shantou University Medical College, Shantou 515041, China; 3Department of Cardiology, Guangdong Cardiovascular Institute, Guangdong Provincial People’s Hospital, Guangdong Academy of Medical Sciences, Guangzhou 510080, China

**Keywords:** coronary hemodynamics, optical coherence tomography, optical flow ratio, quantitative flow ratio, coronary heart disease

## Abstract

Objectives: The study aimed to evaluate the diagnostic performance of optical coherence tomography (OCT) in identifying functionally significant coronary stenosis in a vessel with single stenosis. Background: The OCT-based morphofunctional computational method for deriving the optical flow ratio (OFR) has diagnostic value, as it can identify the functional severity of coronary stenosis, but the ability of the OFR to aid the OCT in determining coronary stenosis hemodynamics in single-stenosis lesion remains unclear. Methods: 74 vessels with single stenosis were studied in 69 patients; all cases were performed through OCT and quantitative flow ratio (QFR), and OCT images were used to perform OFR. Results: Among vessels with single stenosis, OFR showed a good correlation with QFR (r = 0.86; *p* < 0.001). Taking QFR as the standard, the vessel-level diagnosis accuracy, sensitivity, specificity, positive predictive value (PPV) and negative predictive value (NPV) of OFR were 90% (95% CI: 81 to 96), 94% (95% CI: 77 to 99), 88% (95% CI: 74 to 96), 85% (95% CI: 68 to 94) and 95% (95% CI: 82 to 99), respectively. Among vessels with OFR/QFR concordance, both the minimum lumen area (MLA) and minimum lumen diameter (MLD) showed excellent diagnostic efficiency (MLA: area under the curve (AUC) = 0.92, 95% CI: 0.85 to 0.98, *p* < 0.001; MLD: AUC = 0.93, 95% CI: 0.86 to 0.98, *p* < 0.001) in determining the functional significance of coronary stenosis in a single stenosis lesion, and the best cutoff values were 1.55 mm^2^ and 1.40 mm. Conclusions: OFR has a good correlation with QFR. OCT-measured MLA and MLD have excellent diagnostic efficiency in identifying the hemodynamic significance of coronary stenosis in a vessel with single stenosis.

## 1. Introduction

Previous studies have demonstrated that coronary hemodynamics guiding percutaneous coronary intervention (PCI) is superior to that guided by coronary angiography only [1,2,3]. Fractional flow reserve (FFR), a classical tool to evaluate the coronary function, can optimize PCI, reduce the number of stent implantations and reduce adverse cardiovascular events [3]. However, the high cost of the FFR pressure guide wire, complexity of its operation and discomfort of drug-induced dilation limit its performance and development [4]. To overcome these shortcomings, several new techniques have been invented and showed good agreement with FFR [5,6,7,8]. Quantitative flow ratio (QFR), a novel, intracoronary, wire-free technique that can assess the pressure drop in the vessel based on angiography and the flow velocity of the contrast medium, and optical flow ratio (OFR), a novel optical coherence tomography(OCT)-based morphofunctional computational method dubbed the optical flow ratio, are two important examples.

Intracoronary imaging (e.g., OCT, intravascular ultrasound (IVUS)), another way of clarifying the condition of coronary arteries, can identify coronary lesions and better guide stent implantation compared with quantitative coronary angiography (QCA) [9]. Intracoronary imaging and coronary hemodynamics are optical and reliable aids for coronary heart diseases diagnosis and treatment [9,10]. However, in clinical practice, usually only one of them can be performed, due to the limitations of hospital equipment and medical cost.

At present, intracoronary imaging provides information about the morphology details of a coronary (e.g., minimum lumen area (MLA)) and the type of plaque. Doctors then formulate PCI strategies accordingly and select an appropriate balloon and stent [10]. Studies have explored the correlation between intracoronary imaging and coronary hemodynamics, but only a moderate diagnostic efficiency was found in identifying hemodynamically severe coronary stenoses for both OCT and IVUS [8,11,12,13,14]. Inclusion criteria were set regarding the vessel with stenosis severity but not the lesion size. Since the coronary hemodynamics reflect the situation of the whole vessel, while intracoronary imaging only shows the parameter of the most severe cross-section, the number of vessel lesions may influence their correlation. However, little is known about the relationship between the OCT parameters and coronary hemodynamic deficiency in vessels with single stenosis.

The present study aimed to assess the diagnostic efficiency of OFR in identifying coronary hemodynamics to obtain a reliable hemodynamic judgement.

## 2. Materials and Methods

### 2.1. Study Population

This study retrospectively analyzed patients who underwent coronary angiography and OCT analysis for suspected coronary heart disease between October, 2015, and November, 2020. Patients that underwent coronary angiography in a vessel with single stenosis were enrolled in the study. Vessels with a percent diameter stenosis between 30% and 90% in a vessel (≥2 mm) found using QCA, were included. Vessels < 3 mm from the aorta, a left main trunk, bypass graft lesions, a poor-quality coronary angiogram for QFR (for example, foreshortening or overlap of the culprit vessels, insufficient contrast flush, frequent atrial premature or atrial fibrillation) and OCT images of too poor a quality to measure OCT-derived parameters or perform OFR were excluded. This study was approved by the Research Ethics Committee of Guangdong Provincial People’s Hospital, and all the patients were exempt from signing informed consent.

### 2.2. Coronary Angiography and QCA Analysis

Coronary angiography was recorded by a digital subtraction angiography machine (Allura, Philips, Amsterdam, The Netherlands) at 15 frames/s. The field of view (FOV) was 20 cm × 20 cm–22 cm × 22 cm, the matrix was 512 × 512, the tube current was 500–800 mA and the tube voltage was 60–120 kV. The nonionic contrast agent was injected with a forceful and stable high-pressure syringe pump at a rate of approximately 3 mL/s, with a total of 4 mL. A well-trained technician selected the angiogram from an end-diastolic still-frame. QCA software (Beijing Strong Technology Co., Ltd. Beijing, China) was used to analyze the quantitative coronary angiography values. QCA software automatically delineated the lumen contour, manual correction was allowed and catheter calibration was used as the reference standard. An intermediate QCA technician was selected to analyze the QCA; after that, a senior technician verified all data.

### 2.3. QFR Analysis

A QFR analysis was carried out using the Pulse Medical software (Pulse Medical Imaging Technology Shanghai, Shanghai, China). End-diastolic frames were selected by an experienced technician. The target vessel was an automatically identified lumen contour when the stenosis segment was clearly displayed; if there was an error, manual correction was allowed. After that, the proximal and distal normal segments were selected as the proximal and distal reference diameters, respectively. An intermediate QFR technician was selected to analyze the QFR; after that, a senior technician verified all data. Before QFR analysis, the vessel positions were noted to ensure that the QFR, OFR and QCA could be compared at the same site.

### 2.4. OCT Analysis

An OCT image of the coronary artery was obtained using a DragonFly catheter (SJM, Lightlab Imaging Inc., Westford, MA, USA). The blood was removed by the iso-osmolar contrast agent with a high-pressure syringe pump at a rate of 4 mL/s, with a total of 16 mL. The manual and automated pullback OCT speed was 36 mm/s. OCT images were analyzed by Lightlab Imaging software; before data were measured, OCT catheter calibration was applied based on the site preference. OCT-derived parameters contained MLA, minimum lumen diameter (MLD), diameter stenosis (DS) and area stenosis (AS); the distal reference diameter and proximal reference diameter were measured in this study. The cross-section of the OCT images selected for analysis were (1) the cross-section with a minimal lumen area and (2) the proximal and distal reference cross-section, which showed a normal lumen within 10 mm proximal and distal to the minimal lumen area, while the reference cross-section was far from the side branch. The reference lumen area was defined as the (proximal reference lumen area + distal reference lumen area)/2; the reference diameter was defined as the (proximal reference diameter + distal reference diameter)/2. MLA and MLD were measured from the minimal lumen area; AS and DS were calculated by the following formula: (reference lumen area—MLA)/reference lumen·100% and (reference diameter—MLD)/reference diameter·100%.

### 2.5. OFR Measurements

For OFR measurements, OCT images were analyzed by OctPlus software (version 1.0) (Pulse Medical Imaging Technology, Shanghai, China). Technicians selected the OCT images with coronary blood removed by contrast, and the OFR system automatically recognized the lumen contour. This was allowed to manually correct the lumen contour when there was an error. Then, the proximal normal segment and distal normal segment of the vessels were selected as the reference segments. After that, the technician identified the side branch of the coronary and judged whether the lumen contours of ostial side branches were fully displayed; if they were not fully displayed, a manual technician corrected the error. All the coronaries were analyzed by an intermediate OFR technician and then verified by a senior OFR technician. Before OFR analysis, the technicians were instructed to measure the positions of the heart and coronary so that the coronary OFR and QFR could be compared at the same position. Figure 1 shows an example of the OFR and QFR analysis.

### 2.6. Statistical Analysis

Continuous variables were compared with *t*-tests or Mann–Whitney U tests, while categorical variables were compared with Fisher’s exact tests. We used the QFR as the reference standard, and the diagnostic accuracy of OFR was determined by calculating the sensitivity, specificity, positive predictive value (PPV), negative predictive value (NPV), positive likelihood ratio (+LR) and negative likelihood ratio (−LR), as appropriate. The two-sided 95% confidence intervals (CIs) were estimated to obtain the OFR accuracy. When the QFR and OFR values were less than or equal to 0.80, the functional evaluation of coronary stenosis was significant; when the QFR and OFR values were more than 0.80, the functional evaluation of coronary stenosis was nonsignificant. The correlation between OFR and QFR was determined by Spearman’s correlation coefficient (r). The difference between OFR and QFR was reported using Bland–Altman plots. We used the OFR/QFR concordance to define the vessel with the same coronary hemodynamics performed by QFR and OFR; the receiver operating curve (ROC) was used to calculate the area under curve and cutoff value of these parameters, including the lesion length (LL) and DS in QCA and MLA, MLD, DS, AS, distal reference diameter and proximal reference diameter in OCT. All the statistical analyses were performed with MedCalc (version 14.12, MedCalc Software, Ostend, Belgium). A two-sided value of *p* < 0.05 was considered statistically significant.

## 3. Results

### 3.1. Baseline Clinical and Lesion Characteristics

From October 2015 to November 2020, 69 patients with 74 vessels underwent OCT (SJM, Lightlab Imaging Inc., Westford, MA, USA) at Guangdong Provincial People’s Hospital. Seventy-four vessels with a single stenosis were included in this study. The relevant baseline characteristics of the selected patients are presented in Table 1.

The left anterior descending artery (LAD) was the most common vessel (51 (68.92%)). QFR and OFR had mean values of 0.78 ± 0.16 and 0.79 ± 0.15, respectively. QFR ≤ 0.80 was noted in 31 (41.89%) vessels, while OFR ≤ 0.80 was noted in 34 (45.95%) vessels. The mean lesion length (LL) in QCA was 14.87 ± 8.71 mm, and the DS in QCA had a mean value of 52.32 ± 11.23%. Vessels with hemodynamic insufficiency (QFR ≤ 0.80) showed a more severe DS in QCA (58.58 ± 10.03% vs. 47.94 ± 9.97%, *p* < 0.001), while the reference vessel diameter showed no difference between the two groups (2.94 ± 0.54 mm vs. 3.16 ± 0.57 mm, *p* = 0.105) (Table 2).

### 3.2. Agreement between QFR and OFR

In the 74 vessels that underwent coronary hemodynamic assessment by both QFR and OFR, a good correlation was found between OFR and QFR (r = 0.86 (95% CI: 0.78 to 0.91); *p* < 0.001). The Bland–Altman analysis showed no significant difference between OFR and QFR (mean difference = 0.00; SD difference = 0.08; *p* = 0.782), as presented in Figure 2.

### 3.3. Diagnostic Performance of OFR

Using a cutoff value (≤0.80) of QFR to define the functional relevance of stenosis, the vessel diagnostic accuracy of OFR was 90.54% (95% Cl: 80.91 to 95.79), with 38 true positives, 29 true negatives, 2 false positives and 5 false negatives. The vessel-level diagnostic sensitivity, specificity, PPV, NPV, (+) LR and (−) LR of OFR were 93% (95% CI: 77 to 98), 88% (95% CI: 74 to 95), 85% (95% CI: 68 to 94), 95% (95% CI: 81 to 99), 8.04 (95% CI: 3.51 to 18.43) and 0.07 (95% CI: 0.01 to 0.20), respectively (Table 3). Clinical discordance occurred in 7 (9.46%) vessels, with 5 vessels having a QFR > 0.80 but an OFR ≤ 0.80, whereas 2 vessels had a QFR ≤ 0.80 but an OFR > 0.80.

Diagnostic accuracy was defined as the consistency ratio of OFR-evaluated outcomes (≤0.8 or >0.8) to the reference standard QFR-evaluated outcomes (≤0.8 or >0.8). Sensitivity was defined as the proportion of the OFR ≤ 0.8 in vessels with hemodynamically significant stenosis; specificity was defined as the proportion of the OFR >0.8 in vessels without hemodynamically significant stenosis.

PPV = positive predictive value, NPV = negative predictive value, (+) LR = positive likelihood ratio and (−) LR = negative likelihood ratio.

### 3.4. Hemodynamics Performed by QFR and OFR

Despite a good correlation between OFR and QFR and the high diagnostic accuracy of OFR, there were still four lesions whose differences between QFR and OFR were greater than 0.20, and three vessels with a difference between QFR and OFR that was less than or equal to 0.02. The details of these seven vessels are provided in Table 4.

### 3.5. The Correlation between OCT-Derived Intracoronary Stenosis Parameters and Functional Significance of Coronary Stenosis

Considering the difference between OFR and QFR, we analyzed the OCT-derived parameters among the 67 vessels with OFR/QFR concordance. Vessels with a functional significance of coronary stenosis showed a longer lesion length and more severe DS% in QCA compared with those without a functional significance of coronary stenosis (Lesion Length: 27.24 ± 8.99 mm vs. 12.20 ± 5.24 mm, *p* < 0.001; DS%: 58.52 ± 10.38% vs. 47.38 ± 10.19%, *p* < 0.001). The OCT-measurement MLA, MLD, DS% and AS% were more severe in the group with a physiological significance of coronary stenosis (MLA: 1.26 ± 0.45 mm^2^ vs. 2.52 ± 1.04 mm^2^, *p* < 0.001; MLD: 1.25 ± 0.19 mm vs. 1.75 ± 0.34 mm, *p* < 0.001; DS%: 51.02 ± 9.35% vs. 42.01 ± 8.91%, *p* < 0.001; AS%: 75.02 ± 10.18% vs. 65.63 ± 10.14%, *p* < 0.001). The distal and proximal reference diameters were smaller in the hemodynamic significance of the coronary stenosis group (distal reference diameter: 2.40 ± 0.37 mm vs. 2.95 ± 0.45 mm, *p* < 0.001; proximal reference diameter: 2.77 ± 0.37 mm vs. 3.13 ± 0.50 mm, *p* = 0.002) (Table 5).

As shown in Figure 3, the lesion length in QCA, MLA and MLD in OCT showed an excellent predictive value for coronary hemodynamic deficiency (Lesion length: AUC = 0.93, 95% CI: 0.87 to 0.99, *p* < 0.001; MLA: AUC = 0.92, 95% CI: 0.85 to 0.98, *p* < 0.001; MLD: AUC = 0.93, 95% CI: 0.86 to 0.98, *p* < 0.001), and their best cutoff values were 19.19 mm, 1.55 mm^2^ and 1.40 mm, respectively. The remaining factors, DS% in QCA and DS%, AS% and distal and proximal reference diameter in OCT showed a good predictive value (Table 5, Figure 3) 

## 4. Discussion

In this research, we found the following: (1) overall, OFR showed a high diagnostic performance in detecting hemodynamically significant coronary artery disease, as judged by QFR; (2) lesion length >19.19 mm in QCA, MLA ≤ 1.55 mm^2^ and MLD < 1.40 mm in OCT showed an excellent predictive value for the physiological significance of coronary stenosis, as confirmed by the OFR/QFR concordance in vessels with a single stenosis.

### 4.1. OFR with Coronary Hemodynamic Insufficiency

FFR is an important means of evaluating borderline coronary lesions. FAME studies have showed us the advantages of FFR in guiding PCI [8,15,16]. FFR only provides the value of coronary hemodynamics, cannot show detailed anatomical information of vessels and does not objectively identify vessel conditions with high-risk characteristics, such as erosive plaque, thin cap fibro atheroma (TCFA) and the vulnerable plaque of lesions. It has been pointed out that about 20% of lesions with FFR > 0.80 have high-risk characteristics under OCT, such as thin cap fibro atheroma [17,18]. Although TCFA has not yet led to hemodynamics changes, it is still a powerful predictor of major adverse cardiovascular events [17,18]. Therefore, with the aim of optimizing the PCI, intracoronary imaging, such as OCT, with a high resolution, can accurately identify the TCFA [19], which makes up for the FFR deficiencies. OCT obtains the morphological information of coronary plaques through a rapid scan. OFR is a novel method of computational physiology based on optical coherence tomography, which can analyze the OCT imaging. OFR has more than 90% accuracy in diagnosis coronary hemodynamics [20], which is a more comprehensive and objective diagnosis method and helps the doctor to form a PCI strategy. Xu [6] used FFR as the gold standard. The accuracy, sensitivity and specificity of QFR were found to be 92.7%, 94.6% and 91.7%, respectively, in the FAVOR II China study. The accuracy of μQFR (QFR-based Murray law) was 93.0%; the sensitivity and specificity of μQFR was 87.5% and 96.2%, respectively, in the Tu study [21]. In this study, using QFR ≤ 0.80 as a standard, the diagnostic accuracy of OFR for detecting a functionally significant stenosis in coronary artery disease was 90%; the sensitivity and specificity were 87% and 92%, respectively. This was the same as Tu [20] (93%, 92% and 93%).

### 4.2. Difference between OFR and QFR

There were four lesions with a difference between QFR and OFR that was greater than 0.20. This may be explained by the following: (1) when OCT is performed, the vessels may vasospasm due to the OCT catheter, which makes the OFR smaller than the QFR; (2) the OFR analysis has an excellent clarification of the 3D structure of the coronary arteries, while QFR reflects the blood flow better. The difference was less than or equal to 0.02 in three lesions; this difference represents a system error in QFR and OFR when evaluating borderline lesions.

### 4.3. MLA and MLD in OCT with Coronary Hemodynamic Insufficiency

OCT and IVUS are the most widely used intracoronary imaging systems. OCT provides more information for decision-making regarding coronary revascularization [8]. Thus, it is important to obtain information on its potential to identify coronary functional stenosis severity. Previous research focused on the intermediate angiographic severity lesions, but only a moderate value was declared [12,13,22]. The present study evaluated the diagnostic efficiency of OCT parameters in identifying hemodynamically severe coronary stenosis, as determined by the OFR/QFR concordance with vessels with a single stenosis.

In this research, among the six parameters derived from OCT, MLA and MLD had the highest predictive value for hemodynamically significant coronary stenosis. OCT-derived MLA and MLD can help doctors to evaluate coronary stenosis without FFR. Both MLA and MLD have showed an excellent diagnostic ability (MLA: AUC = 0.92; MLD: AUC = 0.93) in detecting coronary hemodynamic deficiency in vessels with single stenosis, while nearly all of the research on vessels with intermediate stenosis only showed a moderate value [8,12,13,23]. The method proved to be a reliable tool for QFR and OFR analyses of coronary hemodynamic evaluation [5,6,7]. Similar to other forms of functional evaluation, they reflect the situation of a length of vessels that may contain and be influenced by two or more lesions. However, MLA and MLD in OCT only provide information for the most severe site. In this research, we evaluated the diagnosis efficiency of OCT parameters in the functional evaluation of vessels with single stenosis and found a high diagnosis value. This reminded us that, in single-stenosis vessels, MLA and MLD may also form a reliable scale for coronary functional deficiencies. However, this conclusion was drawn from a relatively small sample size with tight selection criteria, which limits its expansion and implication during clinical practice. Further research is needed to confirm the diagnostic efficiency of structural parameters for coronary disfunction.

The best cutoff MLA value was 1.55 mm^2^ (vessel diameter < 3 mm) for the hemodynamic dysfunction of coronary arteries, which is much lower than that found in the prior research. Shiono et al. took FFR ≤ 0.75 as the standard, and they found that the best cutoff MLA value in OCT was 1.91 mm^2^ (the sensitivity and specificity were 93.5% and 77.4%), among 62 borderline lesions in 59 patients [24]. Nieve Gonzalo et al. also found that an MLA ≤ 1.95 mm^2^ (the sensitivity was 82%; the specificity was 63%) had a moderate diagnostic efficiency in identifying the functional stenosis significance defined by FFR ≤ 0.80 among vessels with diameters over 3 mm. The optimal cutoff value of MLA was 1.62 mm^2^ (the sensitivity was 80%; the specificity was 83%) in the small vessels’ subgroup (reference diameter < 3 mm) [8]. In the present study, we only included vessels with a single stenosis, which led to a smaller MLA for the same FFR threshold. The populations involved in this research were Chinese, with relatively small reference lumen areas [5]. The MLD in OCT also demonstrated a high hemodynamic prediction value. In this study, the cutoff of the MLD for predicting coronary hemodynamics ≤ 0.80 was 1.40 mm. Fifty-five vessels with a reference diameter of <3 mm were used to detect functionally significant lesions in a Pyxaras study; the best MLD cutoff value in OCT was 1.53 mm (AUC: 0.88, Accuracy: 80%) [25].

Limitations of this study: Firstly, this study was a single-center study, the sample size was small and most of the vessels were LAD. Secondly, a pressure-wire-based FFR, the gold standard for coronary hemodynamics, was lacking in this research, although the diagnostic accuracy of the OFR and QFR was over 90%. Thirdly, the QFR we used was a second-generation QFR without 3D modeling, meaning that the accuracy of the vessel with eccentric plaque is low. Fourthly, there were no follow-up data for patients in the study. Last but not least, despite the tight selection criteria, in 7 of 74 lesions, there was a lack of agreement between the QFR and OFR. This proportion may be greater in the real world.

## 5. Conclusions

The OFR has a good correlation with the QFR. Meanwhile, MLA and MLD in OCT have an excellent diagnostic efficiency in identifying hemodynamically severe coronary stenoses in single lesions, as judged by the OFR and QFR. However, further research is still needed to confirm this result.

## Figures and Tables

**Figure 1 jcm-11-05198-f001:**
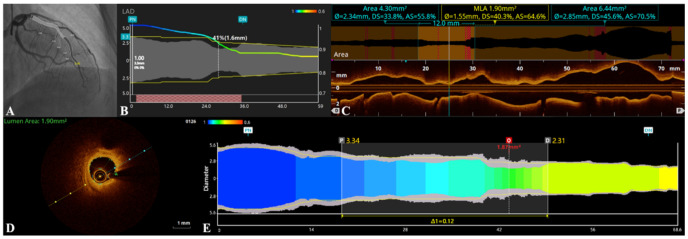
Computation of optical flow ratio (OFR) and quantitative flow ratio (QFR). (**A**) Coronary angiography shows the left anterior descending artery (LAD) with a single stenosis, the QFR value of this LAD was 0.88. (**B**) The virtual lumen and QFR pressure pullback at every position. (**C**) The 75-mm-long OCT pullback done across this LAD; the OCT lumen-Mode and lumen profile was shown, and the diameter of the distal reference and proximal reference measured by the OCT system was 2.34 mm and 2.85 mm, respectively. (**D**) The minimal lumen area (MLA) of LAD was 1.90 mm^2^. (**E**) The OFR software rendered a virtual pressure pullback within this LAD for optimal co-registration between pressure-drop and anatomy; the OFR value of this LAD was 0.86.

**Figure 2 jcm-11-05198-f002:**
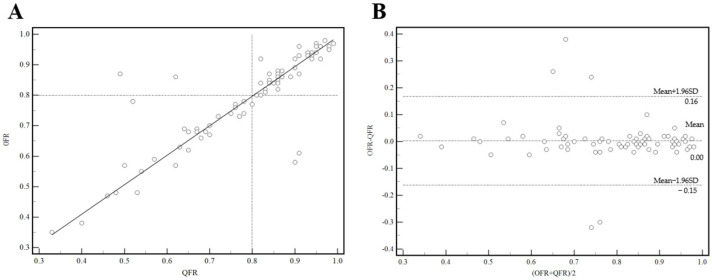
Association between quantitative flow ratio (OFR) and optical flow ratio (QFR). (**A**) Lineal regression between OFR and QFR, (**B**) The Bland-Altman plot presented a difference between the OFR and QFR. SD: standard deviation.

**Figure 3 jcm-11-05198-f003:**
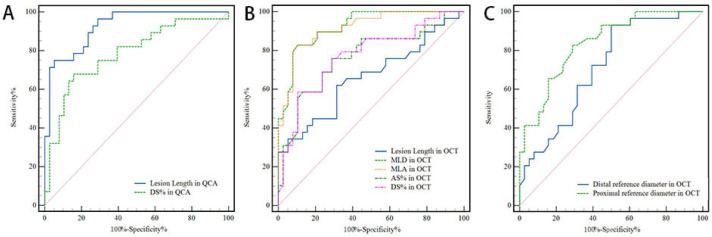
Comparison of receiver operating curves for the discrimination of coronary hemodynamic deficiency. (**A**) Comparison of receiver-operating curves for Lesion length and DS% in QCA; (**B**) Comparison of receiver-operating curves for Lesion length, MLA, MLD, AS%, DS% in OCT; (**C**) Comparison of receiver-operating curves for Distal and proximal reference diameters in OCT. ROC = receiver operating curve, QCA = quantitative coronary angiography, DS% = percent diameter stenosis, OCT = optical coherence tomography, MLA = minimal lumen area, MLD = minimal lumen diameter, AS% = percent area stenosis, OFR = optical flow ratio.

**Table 1 jcm-11-05198-t001:** Baseline clinical characteristics.

	*n* = 69
Age, yrs	63.14 ± 8.57
Female	15 (21.74%)
Left ventricular ejection fraction, %	63.43 ± 6.46
Diabetes mellitus	14 (20.29%)
Hyperlipidemia	3 (4.35%)
Current smoker	18 (26.09%)
Hypertension	23 (33.33%)
Family history of coronary artery disease	2 (2.90%)
Previous myocardial infarction	8 (11.59%)
Stable angina pectoris	59 (85.51%)
Unstable angina pectoris	10 (14.49%)

Continuous values are mean ± SD, Categorical values are *n* (%).

**Table 2 jcm-11-05198-t002:** Vessel characteristics with QFR ≤ 0.80 or >0.80.

	Overall (*n* = 74)	QFR > 0.80 (*n* = 43)	QFR ≤ 0.80 (*n* = 31)	*p*
LAD	51 (68.92%)	25 (58.14%)	26 (83.87%)	0.018
LCX	7 (9.46%)	5 (11.63%)	2 (6.45%)	0.728
RCA	14 (18.92%)	11 (25.58%)	3 (9.68%)	0.085
Diagonal branch	2 (2.70%)	2 (4.65%)	0 (0.00%)	0.506
Lesion length in QCA, mm	14.87 ± 8.71	12.23 ± 5.44	18.64 ± 10.98	0.002
Reference vessel diameter, mm	3.07 ± 0.56	3.16 ± 0.57	2.94 ± 0.54	0.105
Diameter stenosis in QCA, %	52.32 ± 11.23	47.94 ± 9.97	58.58 ± 10.03	<0.001
QFR (per vessel)	0.78 ± 0.16	0.89 ± 0.05	0.63 ± 0.12	<0.001
Vessels with QFR ≤ 0.80	31	0	31	
OFR (per vessel)	0.79 ± 0.15	0.88 ± 0.08	0.66 ± 0.13	<0.001
Vessels with OFR ≤ 0.80	34	5	29	

Continuous values are mean ± SD, Categorical values are *n* (%). LAD = left anterior descending branch, LCX = left circumflex branch, RCA = right coronary artery, QCA = quantitative coronary angiography, QFR = Quantitative flow ratio, OFR = optical flow ratio.

**Table 3 jcm-11-05198-t003:** Diagnostic performance of OFR.

	QFR ≤ 0.80, (95% CI)	No. of Vessels in Group
Accuracy, %	90.54 (80.91–95.79)	74
Sensitivity, %	93.55 (77.16–98.87)	31
Specificity, %	88.37 (74.12–95.64)	43
PPV, %	85.29 (68.17–94.46)	34
NPV, %	95.00 (81.79–99.13)	40
(+) LR	8.04 (3.51–18.43)	
(−) LR	0.07 (0.01–0.20)	

Values are *n* (95% CI) for (+) LR and (−) LR and *n*% (95% CI) for all other parameters.

**Table 4 jcm-11-05198-t004:** The patient’s hemodynamics differences between OFR and QFR.

No	Vessel	Age	Gender	Lesion Length in QCA (mm)	MLA in OCT (mm^2^)	MLD in OCT (mm)	DS% in OCT (%)	AS% in OCT (%)	Distal Reference Diameter in OCT (mm)	Proximal Reference Diameter in OCT (mm)	OFR	QFR
1	LAD	62	M	9.91	1.60	1.42	47.8	72.8	2.46	2.98	0.80	0.82
2	RCA	61	M	14.43	1.69	1.46	49.0	73.8	2.78	2.94	0.86	0.62
3	LAD	58	M	16.96	1.76	1.49	50.4	75.2	2.95	3.06	0.87	0.49
4	LAD	65	F	25.82	1.63	1.42	46.2	70.7	2.42	2.86	0.80	0.82
5	LAD	61	M	22.35	1.48	1.37	40.2	64.2	2.28	2.30	0.61	0.91
6	LAD	57	M	20.35	1.02	1.14	44.7	70.2	1.75	2.37	0.58	0.90
7	LAD	44	M	7.81	1.11	1.19	62.6	86.1	3.18	3.19	0.80	0.81

LAD = left anterior descending branch, RCA = right coronary artery, QCA = quantitative coronary angiography, MLA = minimum lumen area, OCT = optical coherence tomography, MLD = minimum lumen diameter, DS% = diameter stenosis, AS% = area stenosis, OFR = optical flow ratio, QFR = quantitative flow ratio, M = male, F = female.

**Table 5 jcm-11-05198-t005:** Differences between parameters in QCA and OCT and the area of ROC and Cut-off value of parameters in QCA and OCT with OFR/QFR concordance.

	QFR ≤ 0.80 and OFR ≤ 0.80 (*n* = 38)	QFR > 0.80 and OFR > 0.80 (*n* = 29)	*p*	AUC (95%CI)	Cut off Value
QCA-based characteristics					
Lesion length (mm)	27.24 ± 8.99	12.20 ± 5.24	<0.001	0.93(0.87–0.99)	19.19
DS (%)	58.52 ± 10.38	47.38 ± 10.19	<0.001	0.78(0.67–0.90)	57.52
OCT-based characteristics					
MLA (mm^2^)	1.26 ± 0.45	2.52 ± 1.04	<0.001	0.92(0.85–0.98)	1.55
MLD (mm)	1.25 ± 0.19	1.75 ± 0.34	<0.001	0.93(0.86–0.98)	1.40
DS% (%)	51.02 ± 9.35	42.01 ± 8.91	<0.001	0.77(0.65–0.89)	51.65
AS% (%)	75.02 ± 10.18	65.63 ± 10.14	<0.001	0.76(0.64–0.88)	70.45
Distal reference diameter (mm)	2.40 ± 0.37	2.95 ± 0.45	<0.001	0.83(0.74–0.93)	2.70
Proximal reference diameter (mm)	2.77 ± 0.37	3.13 ± 0.50	0.002	0.71(0.59–0.83)	3.20

QCA = quantitative coronary angiography, DS% = percent diameter stenosis, OCT = optical coherence tomography, MLA = minimum lumen area, MLD = minimum lumen diameter, AS% = percent area stenosis, OFR = optical flow ratio.

## Data Availability

The data presented in this study are available on request from the corresponding author. The data are not publicly available due to privacy issues.

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
