# Peer review of "Morphometric Assessment for Functional Evaluation of Coronary Stenosis with Optical Coherence Tomography and the Optical Flow Ratio in a Vessel with Single Stenosis"

_jcm, 2022, doi:10.3390/jcm11175198_

Round 1
Reviewer 1 Report
Yuming Huang et al retrospectively evaluated the correlation between QFR and OFR in patients undergoing OCT for single lesions. They also aimed to identify OCT derived parameters that might predict hemodynamically significant lesions. The manuscript is of interest as there are not a lot of data regarding the performance of OFR.
1. Please elaborate more in the methods and discussion regarding the QFR and OFR techniques (which part of the artery is being evaluated in each technique, is the entire coronary tree anatomy taken into consideration? Are there any projections that are required for QFR? what are the advantages and disadvantages of each technique, Etc)
2. I'm not sure I understand the purpose of section 3.4 in the results – the findings presented in Table 4 are not discussed in the discussion section- what do we learn from the cases in which QFR and OFR were nor concordant?
3. Section 3.5 and table 5 are not really clear to me. It is already known that lesions that are longer and of higher severity are more likely to be hemodynamically significant- I'm not sure what is the contribution of this section (in the results and discussion) to the manuscript
Also, the authors write: "Considering about the difference between OFR and QFR, we analyzed the relation between OCT-derived parameters among the 67 vessels with OFR/QFR concordance and 7 vessels with OFR/QFR discordance" however the comparison in table 5 is between QFR ≤0.80 And OFR ≤ 0.80 versus QFR >0.80 And OFR > 0.80
4. I think the discussion section should focus on previous data regarding the performance of OFR, previous studies comparing OFR versus QFR and FFR etc, instead of looking again at lesion characteristics. We all know that the hemodynamic significance of a lesion depend on many factors- the territory that is being supplied by the artery, focal versus diffuse disease, big versus small vessels. OCT parameters alone do not take into consideration all these factors
Author Response
Dear Editors and Reviewers:
Thank you for your letter and for the reviewers’ comments concerning our manuscript entitled “Morphometric assessment for functional evaluation of coronary stenosis with optical coherence tomography and optical flow ratio in vessel with single stenosis" (ID: jcm-1848691). Those comments are all valuable and very helpful for revising and improving our paper, as well as the important guiding significance to our researches. We have studied comments carefully and have made correction which we hope meet with approval. The main corrections in the paper and the responds to the reviewer’s comments are as following:
Responds to the reviewer’s comments:
- Response to comment: (- Please elaborate more in the methods and discussion regarding the QFR and OFR techniques (which part of the artery is being evaluated in each technique, is the entire coronary tree anatomy taken into consideration? Are there any projections that are required for QFR? what are the advantages and disadvantages of each technique, Etc) )
Respond: thanks for the reviewer’s suggestion. In QFR and OFR calculation, not the entire coronary tree anatomy is taken into consideration, but as long as possible the length of the target vessel is enrolled into measurement. For QFR analysis, Murray-law based quantitate flow ratio (µQFR) was used in this research, thus, only a single angiographic projection was needed during analysis. The advantages and disadvantages of QFR and OFR were further discussed in the discussion section (page 8-9, line 316-323).
- Response to comment: (-I'm not sure I understand the purpose of section 3.4 in the results – the findings presented in Table 4 are not discussed in the discussion section- what do we learn from the cases in which QFR and OFR were nor concordant?)
Respond: thanks for the reviewer’s comments. It is true that we are lacking a discussion on the result of section 3.4. and table 4. We have added “4.2. Difference between OFR and QFR” in the discussion section and discusses about the clinical and technical differences of QFR and OFR measurement (page 8-9, line 316-323).
- Response to comment: (-Section 3.5 and table 5 are not really clear to me. It is already known that lesions that are longer and of higher severity are more likely to be hemodynamically significant- I'm not sure what is the contribution of this section (in the results and discussion) to the manuscript)
Respond: thanks for the review’s comments. We are sorry that we did not make it clear in this section. The quoted parted was altered into: “Considering about the difference between OFR and QFR, we analyzed the OCT-derived parameters among the 67 vessels with OFR/QFR concordance.” (page 7, line 237-238) and hope it makes a better announcement.
- Response to comment: (-I think the discussion section should focus on previous data regarding the performance of OFR, previous studies comparing OFR versus QFR and FFR etc, instead of looking again at lesion characteristics. We all know that the hemodynamic significance of a lesion depended on many factors- the territory that is being supplied by the artery, focal versus diffuse disease, big versus small vessels. OCT parameters alone do not take into consideration all these factors)
Respond: thanks to reviewer’s suggestion, in the discussion section, further discussion about previous studies comparing OFR versus QFR and FFR are added to complete it (page 8, line 308-312).
We appreciate for Editors/Reviewers’ warm work earnestly, and hope that the correction will meet with approval.
Once again, thank you very much for your comments and suggestions. We look forward to hearing from you.
Sincerely
Reviewer 2 Report
In this retrospective study, the authors evaluated the diagnostic performance of optical coherence tomography (OCT) in the identification of functionally significant coronary stenosis in vessels with single stenosis. To this aim, optical flow ratio (OFR), as a novel OCT-based computational technique, was used and compared with quantitative flow ratio (QFR). The authors reported that OFR had a high diagnostic performance in detecting hemodynamically significant coronary artery disease, as judged by QFR. In addition, in this study lesion length >19.19 mm in QCA, MLA ≤1.55 mm² and MLD <1.40 mm in OCT had an excellent predictive value for the physiological significance of coronary stenosis.
This study addresses a current topic as it compares an OCT-based computational technique (OFR) with an angiography-based computational method (QFR) in the assessment of a functionally significant coronary stenosis in vessels with single stenosis.
1. In the introduction, the authors should briefly explain the principles and relevance of OFR and QFR in the evaluation of coronary stenosis. Furthermore, the authors should briefly comment on previous research reporting associations of stenosis geometry with QFR (e.g., Milzi JCM 2021).
2. An important aspect of this study is that left anterior descending artery (LAD) was the predominantly analyzed vessel. The authors should explain the impact of this aspect on the evaluation of the diagnostic performance of OFR in other coronary vessels in the present study.
3. In a total of 7 vessels, the authors found differences between OFR and QFR. The authors should elucidate possible explanations for this observation.
4. In this study, the authors reported an excellent predictive value of OCT-derived MLA and MLD for the hemodynamic significance of coronary stenosis. Why is this result different from previously published data reporting only a moderate association? May this difference represent a bias in evaluating OFR?
5. The authors should check the manuscript for accuracy and mistakes in writing.
Author Response
Dear Editors and Reviewers:
Thank you for your letter and for the reviewers’ comments concerning our manuscript entitled “Morphometric assessment for functional evaluation of coronary stenosis with optical coherence tomography and optical flow ratio in vessel with single stenosis" (ID: jcm-1848691). Those comments are all valuable and very helpful for revising and improving our paper, as well as the important guiding significance to our researches. We have studied comments carefully and have made correction which we hope meet with approval. The main corrections in the paper and the responds to the reviewer’s comments are as following:
Responds to the reviewer’s comments:
- Response to comment: (In the introduction, the authors should briefly explain the principles and relevance of OFR and QFR in the evaluation of coronary stenosis. Furthermore, the authors should briefly comment on previous research reporting associations of stenosis geometry with QFR (e.g., Milzi JCM 2021).)
Respond: thanks to reviewer’s suggestion, we briefly explain the principles and relevance of OFR and QFR in the introduction section(page 2, line 45-48) and a discussion on previous report was also added(page 2, line 59-61,reference 15) .
- Response to comment: (An important aspect of this study is that left anterior descending artery (LAD) was the predominantly analyzed vessel. The authors should explain the impact of this aspect on the evaluation of the diagnostic performance of OFR in other coronary vessels in the present study.).
Respond: we agree to the reviewer’s comments, but it is a common condition nearly in all the similar articles and hard to be solved, so we stressed it as a limitation in our discussion part (page 9, line 362-363)
- Response to comment: (In a total of 7 vessels, the authors found differences between OFR and QFR. The authors should elucidate possible explanations for this observation.)
Respond: we appreciated the suggestions of the reviewer. A further discussion about these 7 vessels were added in the discussion section (page 8-9, line 316-323).
- Response to comment: (In this study, the authors reported an excellent predictive value of OCT-derived MLA and MLD for the hemodynamic significance of coronary stenosis. Why is this result different from previously published data reporting only a moderate association? May this difference represent a bias in evaluating OFR?)
Respond: thanks to reviewer’s suggestion, in the previous researches, researchers include vessels with intermediate stenosis but not consider about the amount of lesion. In this research, only vessel with single stenosis were included, we believe that it is the reason why we found an excellent predictive value, but the sample of this research is relative small, thus a bias in OFR evaluating can not be totally excluded.
- Response to comment: (The authors should check the manuscript for accuracy and mistakes in writing.)
Respond: thanks to reviewer’s suggestion, and we ask the polish of the content by native speaker, which the modification was trackable in our submitted revised documents.
We appreciate for Editors/Reviewers’ warm work earnestly, and hope that the correction will meet with approval.
Once again, thank you very much for your comments and suggestions. We look forward to hearing from you.
Sincerely